# Urgency urinary incontinence, loss of independence, and increased mortality in older adults: A cohort study

Takashi Yoshioka[1,2], Tsukasa Kamitani[2,3], Kenji Omae[1,2,4], Sayaka Shimizu[3,5], Shunichi Fukuhara[1,5], Yosuke Yamamoto[2]*

1 Center for Innovative Research for Communities and Clinical Excellence (CiRC2LE), Fukushima Medical University, Fukushima, Japan, 2 Department of Healthcare Epidemiology, Kyoto University School of Public Health in the Graduate School of Medicine, Kyoto, Japan, 3 Institute for Health Outcome & Process Evaluation Research (iHope international), Kyoto, Japan, 4 Department of Innovative Research & Education for Clinicians & Trainees (DiRECT), Fukushima Medical University, Fukushima, Japan, 5 Section of Clinical Epidemiology, Department of Community Medicine, Kyoto University, Kyoto, Japan

* yamamoto.yosuke.5n@kyoto-u.ac.jp

## Abstract

### Objectives

To investigate the longitudinal association of urgency urinary incontinence (UUI) with loss of independence (LOI) or death among independent community-dwelling older adults.

### Design

Population-based cohort study.

### Setting

The Locomotive Syndrome and Health Outcome in Aizu Cohort Study (LOHAS), Minami-Aizu Town and Tadami Town, Fukushima, Japan.

### Participants

A total of 1,580 participants aged ≥65 years who underwent a health check-up conducted by LOHAS in 2010.

### Measurements

Exposure was defined as the presence of UUI, which was measured by a questionnaire based on the definition of UUI from the International Continence Society. The primary outcome was defined as incidence of LOI or death. After the check-up in 2010, the outcome was monitored until March 2014. A multivariable Cox proportional hazard analysis was performed to estimate the hazard ratio for the outcome. Ten potential confounders were adjusted in the analysis. Furthermore, we defined the secondary outcomes as two separate outcomes, LOI and death, and performed the same analysis.

**Data Availability Statement:** The data was obtained from the collaboration with the local government and contains sensitive information on individuals including gender, age, and self-reported

social data, and sharing these data openly is prohibited by the contract with the local government. Data requests may be sent to iHope international (research@i-hope.jp). The authors did not have any special privileges to access the data and others would be able to access these data in the same manner as the authors.

**Funding:** The authors received no specific funding for this work.

**Competing interests:** The authors have declared that no competing interests exist.

## Results

Among all participants, 328 reported UUI. The incidence rates of the outcome were 20.4 and 11.4 (per 1,000 person–years) among participants with and without UUI, respectively. After multivariable adjustment, those who experienced UUI showed a substantial association with LOI or death (HR, 1.65; 95% CI, 1.01–2.68). However, they did not show such an association with LOI alone (HR, 1.07; 95% CI, 0.49–2.33). On the other hand, those with UUI exhibited a substantial association with death (HR, 2.23; 95% CI, 1.22–4.31).

## Conclusions

In this study, UUI was associated with the occurrence of LOI or death; however, UUI is not associated with the occurrence of LOI alone among independent community-dwelling older adults. Our results suggest that there may be a difference between UUI-associated diseases that cause LOI and those that cause death.

## Introduction

Loss of independence (LOI), which comprises difficulties in performing activities essential to independent living, are common health burdens among older adults in the aging global population [1,2]. LOI is strongly associated with increased healthcare costs, hospitalization, and increased mortality [1,2]. Therefore, exploring the risk factors of LOI and its prevention are critical issues for healthcare professionals in super-aging societies such as Japan [1,3].

Urinary incontinence (UI) is another health burden faced by aging societies; this condition influences people's health-related quality of life and their physical and social well-being [4,5]. UI is highly prevalent and takes a number of forms; in 2008, urgency UI (UUI), stress UI (SUI]), and mixed UI (MUI) were found to be prevalent in 53.4% of women and 15.1% of men. According to the National Health and Nutrition Examination Survey in the United States, the trend is increasing [6]. Additionally, UI represents a large economic burden associated with management costs and attributed nursing costs, which were estimated to equal 32 billion US Dollars in 2000 [7]. Therefore, early detection, prevention, and appropriate treatments of UI is an essential issue for healthcare professionals worldwide.

The three subtypes of UI—UUI, SUI, and MUI—have different risk factors and etiologies [4]. Previous studies have shown an association between UI, LOI, and the combined outcome of LOI and/or death; however, all of those studies combined the subtypes of UI [8,9]. Other research has suggested that UUI affects falls and mobility impairment, while SUI does not [10,11]. Falls and loss of mobility can lead to LOI and/or death [12,13]. Therefore, we focused on UUI—a heavier burden than other subtypes of UI among geriatric populations—and hypothesized that UUI is a risk factor of both LOI and death among independent older adults. Hence, we aimed to longitudinally investigate the possible associations between UUI and the occurrence of LOI or death among community-dwelling independent older adults using a general cohort in Japan.

## Methods

### Study design and setting

This research was a population-based cohort study using data from the Locomotive Syndrome and Health Outcomes in Aizu Cohort Study (LOHAS) [14]. The LOHAS was conducted at Minami-Aizu and Tadami, in Fukushima, Japan [14].

## Inclusion and exclusion criteria

This study included older adults aged 65 years and older who had undergone a health examination conducted by LOHAS in 2010. Participants with missing UUI data and covariates were excluded from this study. Those with LOI at baseline—defined as care levels three to five by Japan's long-term care insurance (LTCI) certification status—were also excluded [15].

## Measurement of exposure

Exposure was defined as the presence of UUI, which was measured via questionnaire based on the definition of UUI according to the International Continence Society; the overactive bladder (OAB) symptom score has been verified for its applicability in Japanese [16,17]. The questionnaire asked, "How often do you urinate because you cannot defer the sudden desire to urinate? (*never*, *less than once a week*, *once a week or more*, *about once a day*, *2–4 times a day*, and *5 times a day or more*)." The presence of UUI was detected as any UUI response other than "never." We also classified UUI into two categories based on severity: moderate-to-severe UUI (from *less than once a week* to *once a week or more*) and severe UUI (*once a day or more*). This definition has also been used by previous researchers [9,18].

## Main outcome measures

First, the occurrence of LOI or death was defined as a primary outcome. As described above, LOI was determined for those participants at care levels three to five, as designated by the LTCI [15]. All Japanese adults 65 years and older are eligible to apply for LTCI certification, based on physical and/or mental disabilities. To obtain LTCI certification status, authorized care managers and family doctors use standardized questionnaires to evaluate applicants electronically, and automatically estimate their care level. The designated LTCI certification status is then approved by a certification board, which includes individuals from various healthcare professions. The LTCI certification status consists of five levels: levels one and two indicate a person needing only a limited amount of assistance in their basic activities of daily living (ADL); for example, they may require assistance when bathing. Levels three through five indicate complete dependence and a need for assistance for most ADL [15]. Second, the occurrences of LOI and death were defined as two distinct secondary outcomes. LTCI certification status data, care levels, death, and the dates of those events/determinations were obtained from each town, up to March 31, 2014. The outcomes were evaluated from the date of baseline UUI measurement to the end of March, 2014. All participants were monitored until LTCI certification. Monitoring of participants was discontinued if they emigrated from the included towns, withdrew their consent for participation, were no longer being observed by LOHAS, or upon their death.

## Covariates

Ten covariates—age, sex, body mass index (BMI), smoking status, alcohol use, hypertension, dyslipidemia, diabetes mellitus, and histories of heart disease and stroke—were selected as potential confounders. Of these, nine covariates (excluding BMI) were obtained from the questionnaire. BMI was calculated based on actual measurements of weight (kg) and height (cm) at the time of the physical examination.

## Statistical analysis

First, baseline characteristics were described as means (SD) for continuous variables and as frequencies (percentage) for categorical variables. The survival curves were then calculated

using the Kaplan–Meier method. The time calculated for the risk of outcome was from the time participants attended the LOHAS health examination to the occurrence of LOI, death, or the end of the study. We performed multivariable analyses using Cox proportional hazards models to estimate the HRs and 95% CIs for the occurrence of LOI or death (the primary outcome), LOI alone, and death alone (the secondary outcomes) in older adults, with or without UUI. The proportional hazard assumption was assessed by Schoenfeld's residual test and a log-minus-log-survival plot. As the occurrence of death eliminated the probability of the occurrence of LOI, both were modeled with the proportional subdistribution hazard regression described by Fine and Gray [19]. We estimated the subdistribution HR (SHRs) and 95% CIs for the occurrences of LOI and death in the secondary analysis. In the tertiary analysis, the definition of exposure changed from the presence of UUI to the presence of mild-to-moderate UUI and severe UUI, compared with no UUI, to assess whether the results increased with UUI severity. The tertiary analysis followed the same approach as the primary and the secondary analyses. In all three analyses, the above covariates were adjusted for complete case analysis.

The consistency and statistical robustness of the results were confirmed with two sensitivity analyses. The first analysis followed the same analytical approach as the primary and secondary analyses, iterating dataset imputation 20 times (multiple imputations) for the missing covariates [20]. To account for potential model instability due to an insufficient number of secondary outcomes, we created four statistical models for the secondary sensitivity analyses. The four models used were Model 1: crude model, exposure only; Model 2: age and sex, in addition to Model 1; Model 3: BMI, smoking status, and alcohol use, in addition to Model 2; and Model 4: hypertension, dyslipidemia, diabetes mellitus, and histories of heart disease and stroke, in addition to Model 3. STATA version 15.1 (Stata Corp., College Station, TX) software was used for statistical analyses. Statistical significance for Schoenfeld residual test was set at $p > 0.05$.

### Ethical considerations

This study was approved by the institutional review board of Kyoto University and Fukushima Medical University; and followed STROBE guidelines. Written informed consent was obtained from all participants.

## Results

Fig 1 shows a diagram of participant flow. Of the 1,757 participants aged 65 years and older who underwent the medical examination in 2010; seven were excluded because of missing exposure data. An additional 170 participants were excluded due to missing covariate data. No participants were certified as LTCI level three to five at baseline. Finally, 1,580 participants were included in the primary analysis and 1,750 participants were used for the sensitivity analysis after multiple imputations for missing data. The median follow-up period was 1,408 days (interquartile range 1,396–1,416). Eight participants had emigrated and were censored.

Table 1 shows the participant characteristics. The mean total participant age was 72.8±4.7 years, 680 were men (43.0%), and their average BMI was 23.8±3.0 kg/m$^2$. A total of 328 (20.8%) participants reported UUI, compared with participants without UUI (1,252, 79.2%). The characteristics based on UUI severity are shown in S1 Table.

Table 2 details the outcome incidences. The incidence rate of LOI or death among those with or without UUI was 20.4 and 11.4 per 1,000 person–years (95% CI, 13.8–30.2 and 8.7–14.9), respectively. The incidence rates of LOI among those with or without UUI were 7.4 and 5.5 per 1,000 person–years (95% CI, 3.8–14.1 and 3.7–8.1), whereas those of death among those with or without UUI were 13.1 and 5.9 per 1,000 person–years (95% CI, 8.0–21.3 and 4.1–8.5), respectively.

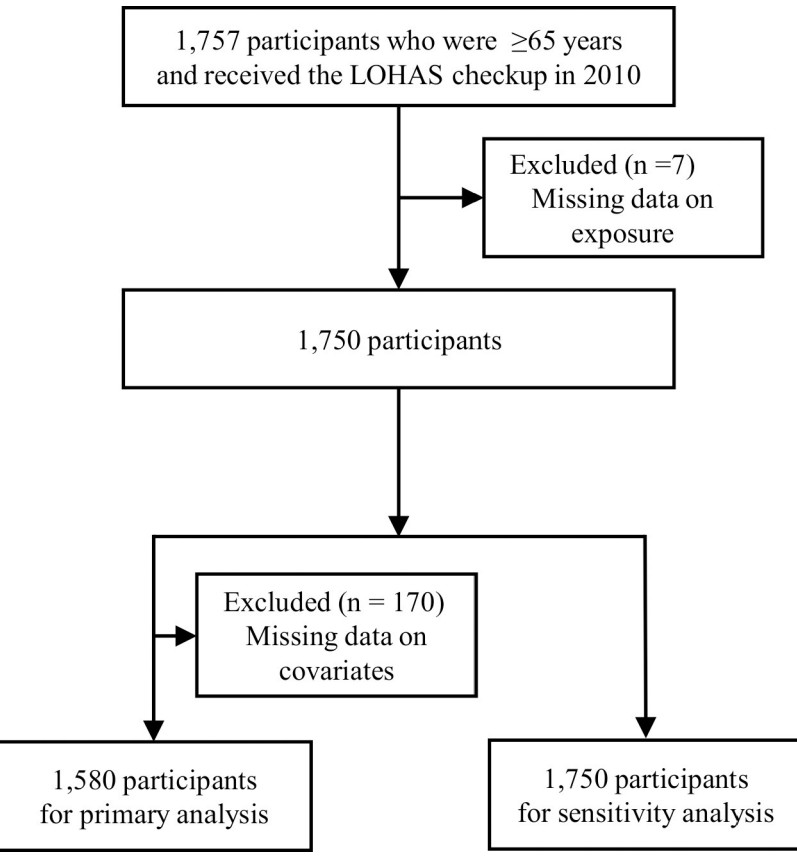

**Fig 1. Participant flow.**

Fig 2 illustrates the survival curve representing the association between UUI and the primary and secondary outcomes using the Kaplan–Meier method (A for LOI or death; B for LOI; C for death). The multivariable Cox proportional hazard analysis indicated that the

**Table 1. Baseline characteristics of participants.**

| | Total | | Participants without UUI | | Participants with UUI | |
|---|---|---|---|---|---|---|
| | *N* = 1,580 | | *n* = 1,252 | | *n* = 328 | |
| | Summary | | Summary | | Summary | |
| Age in years; mean (SD) | 72.8 (±4.7) | | 72.5 (±4.7) | | 74.2 (±4.5) | |
| Sex (male; *n*, %) | 680 | (43.0) | 570 | (45.5) | 110 | 33.5 |
| Body mass index as kg/m²; mean (SD) | 23.8 (±3.0) | | 23.7 (±2.9) | | 24.2 (±3.2) | |
| Smoking status (current smoker; *n*, %) | 130 | 8.2 | 114 | 9.1 | 16 | 4.9 |
| Alcohol use (yes; *n*, %) | 636 | 40.3 | 518 | 41.4 | 118 | 36.0 |
| Hypertension (yes; *n*, %) | 864 | 54.7 | 686 | 54.8 | 178 | 54.3 |
| Dyslipidemia (yes; *n*, %) | 462 | 29.2 | 357 | 28.5 | 105 | 32.0 |
| Diabetes (yes; *n*, %) | 133 | 8.4 | 103 | 8.2 | 30 | 9.1 |
| History of heart disease (yes; *n*, %) | 151 | 9.6 | 113 | 9.0 | 38 | 11.6 |
| History of stroke (yes; *n*, %) | 60 | 3.8 | 45 | 3.6 | 15 | 4.6 |

Note. UUI: urgency urinary incontinence. SD: standard deviation.

**Table 2. The incidence of loss of independence and death.**

|  | Subjects | LOI or death | | LOI | | Death | |
|---|---|---|---|---|---|---|---|
|  | *N* | *n* (%) | incidence rate (95% CI) | *n* (%) | incidence rate (95% CI) | *n* (%) | incidence rate (95% CI) |
| **Overall** | 1,580 | 79 (5.0) | 13.2 (10.6–16.5) | 35 (2.2) | 5.9 (4.2–8.2) | 44 (2.8) | 7.4 (5.5–.9) |
| **No UUI** | 1,252 | 54 (4.3) | 11.4 (8.7–4.9) | 26 (2.1) | 5.5 (3.7–8.1) | 28 (2.2) | 5.9 (4.1–8.5) |
| **Presence of UUI** | 328 | 25 (7.6) | 20.4 (13.8–30.2) | 9 (2.7) | 7.4 (3.8–14.1) | 16 (4.9) | 13.1 (8.0–21.3) |
| Mild-to-moderate UUI | 252 | 16 (6.3) | 16.9 (10.4–27.6) | 7 (2.8) | 7.4 (3.5–15.5) | 9 (3.6) | 9.5 (5.0–18.3) |
| Severe UUI | 76 | 9 (11.8) | 32.4 (16.9–62.3) | 2 (2.6) | 7.2 (1.8–28.8) | 7 (9.2) | 25.2 (12.0–52.9) |

Note. UUI: urgency urinary incontinence. LOI: loss of independence. CI: confidence interval.

presence of UUI was substantially associated with the occurrence of LOI or death (HR, 1.65; 95% CI, 1.01–2.68). On the other hand, the presence of UUI was not substantially associated with the occurrence of LOI (HR, 1.07; 95% CI, 0.49–2.33), and substantially associated with the occurrence of death (HR, 2.29; 95% CI, 1.22–4.31). In all analyses, proportional hazard assumptions were not violated (Schoenfeld global test: $P = 0.40$, $P = 0.89$, and $P = 0.26$; log-minus-log-survival plots are shown in S1 Fig). In the secondary analyses, similar SHRs and cumulative incidences were observed, as shown in S2 Table and S2 Fig. In the tertiary analysis, participants with severe UUI showed a substantial association with the occurrences of LOI or death and death alone, among those with severe UUI (HR, 2.53; 95% CI, 1.23–5.20; and HR, 4.18; 95% CI, 1.78–9.79, respectively). However, there was no substantial association between severe UUI and the occurrence of LOI alone (HR, 1.05; 95% CI, 0.24–4.49), compared with those with no UUI. Among those with mild-to-moderate UUI, no substantial associations with LOI or death, LOI alone, and death alone were observed, compared to those with no UUI (HR, 1.37; 95% CI, 0.78–2.43 for the occurrence of LOI or death; HR, 1.08; 95% CI, 0.46–2.54 for the occurrence of LOI; HR, 1.69; 95% CI, 0.78–3.64 for the occurrence of death). Regarding the occurrences of LOI and death and death alone, the HRs increased as the severity of UUI increased. The Kaplan–Meier survival curves in the tertiary analysis are described in Fig 3. In the tertiary analysis, similar SHRs and cumulative incidences were observed (S2 Table and S2 Fig), and proportional hazard assumptions were not violated (S3 Fig).

In the first sensitivity analysis after multiple imputations, the same associations were observed for any UUI (HR, 1.70; 95% CI, 1.06–2.72 for the occurrence of LOI or death; HR, 1.09; 95% CI, 0.50–2.37 for the occurrence of LOI alone; and HR, 2.29; 95% CI, 1.26–4.17 for the occurrence of death alone). Regarding the second sensitivity analysis, all four statistical models showed consistently similar estimations, as shown in Table 3.

## Discussion

We found an association between UUI and LOI or death after adjusting for ten potential confounders: age, sex, BMI, smoking status, alcohol use, hypertension, dyslipidemia, diabetes mellitus, and histories of heart disease and stroke. We also found a substantial association with death alone; however, we did not find such an association with LOI alone. The HRs of LOI or death and death alone increased with the severity of UUI from *less than once a week* to *once a day or more*. The results were consistent, even after multiple imputations and different covariate selection models. The presence of UUI was 20.8% in this study, which is comparable to other studies that examined individuals aged 65 years and older [18].

Our results suggest that UUI is associated with the occurrence of increased mortality, but not LOI, among independent community-dwelling older adults. Although several previous studies revealed an association between nocturia and LOI or death [21,22], it was unclear

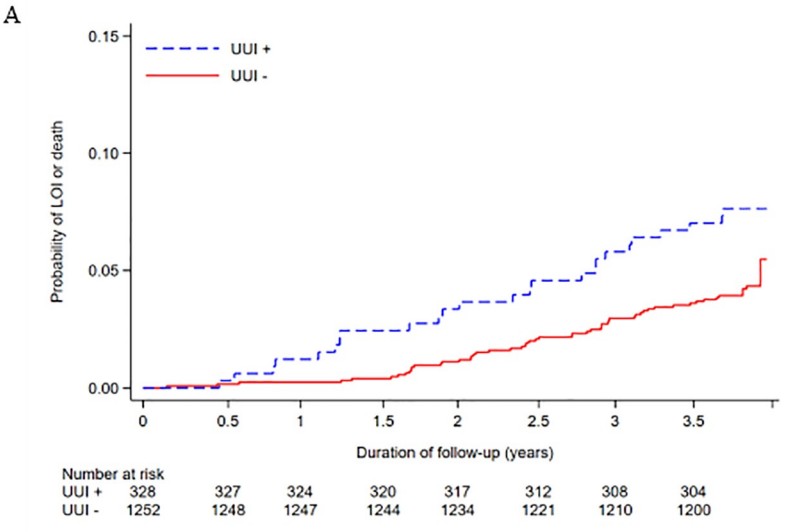

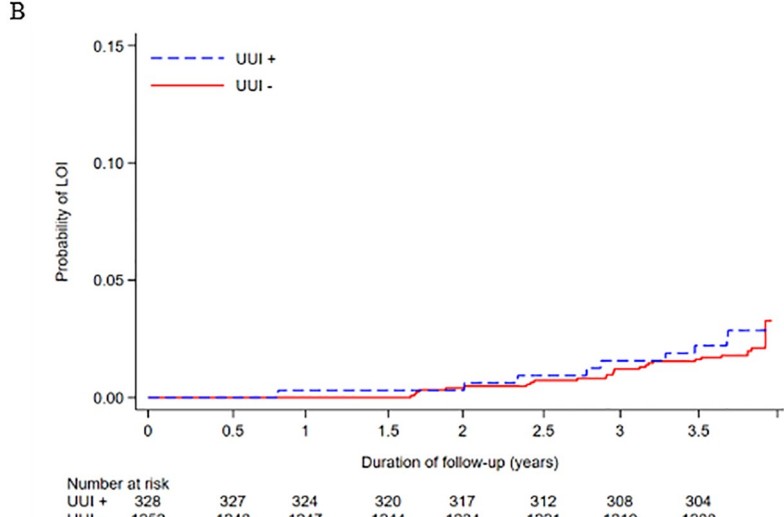

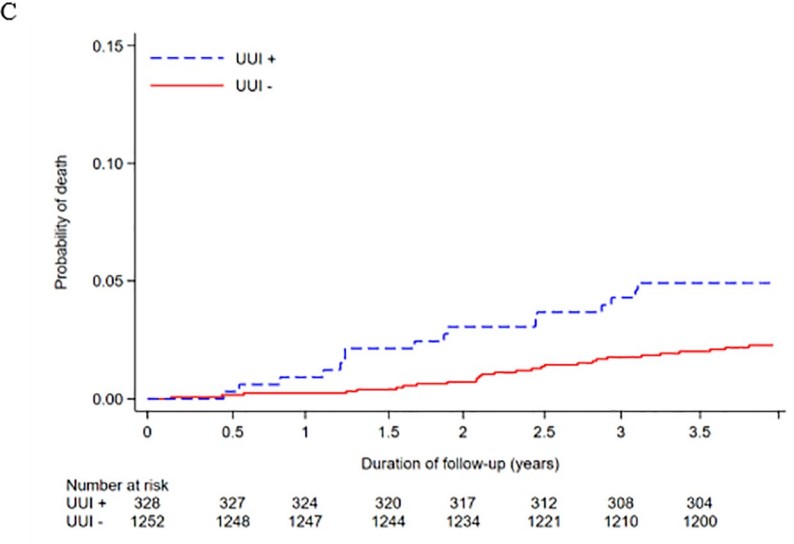

**Fig 2. Survival curve describing the association between UUI and the outcomes using Kaplan–Meier method.** A, LOI or death; B, LOI; C, death.

whether such an association presented for UUI. For example, some studies showed that UUI (not SUI) affected falls, fractures, and mobility impairment, [10,11] evidence illustrating the association between UUI and LOI is scarce. One study indicated that the presence of UI was significantly associated with increased loss of ADL, mobility, and functional limitations among community-dwelling women in the United States (adjusted ORs and 95% CIs: 1.96, 1.07–3.58; 1.26, 1.01–1.56; and 1.36, 1.07–1.73, respectively) [23]. However, their study was limited by it only including women, their cross-sectional study design, and not clarifying the subtypes of UI. In contrast, numerous studies have examined the association between UI and increased mortality. One systematic review observed a significant association between UI and increased mortality (unadjusted HR, 2.22; 95% CI, 1.77–2.78; adjusted HR, 1.27; 95% CI, 1.13–1.42) [9]. Furthermore, pooled HR increased with UI frequency (unadjusted HR for severe UI, 2.27; 95% CI, 1.90–3.87; adjusted HR for severe UI, 1.47; 95% CI, 1.03–2.10). However, the systematic review had a major limitation: it failed to clarify the type of UI (i.e., UUI, SUI, or mixed UI) that influences mortality. To compensate for these limitations, the current study focused on UUI, which is a more relevant condition among geriatric populations and has a heavier burden than SUI [24]. Additionally, this study included general community-dwelling older adults, including men. To the authors' knowledge, this is the first study to investigate the longitudinal association between the presence of UUI and the occurrence of LOI and death.

Our results were similar to previous studies regarding death; however, these results were not comparable to previous studies regarding the occurrence of LOI. Because the association of UUI with increased mortality is considered multifactorial [9], several mechanisms may explain the current results. One explanation is that UUI may reflect systemic pathophysiological changes. A main cause of UUI is OAB, which is characterized by detrusor overactivity [25], and a recent study suggests that detrusor overactivity may be caused by reduced blood flow to the bladder due to atherosclerosis [26]. Another study has suggested that peri-bladder atherosclerosis may be a component of systemic atherosclerosis [27]. It was recently established that lower vitamin D levels are associated with OAB and lower urinary tract symptoms in men [28,29]. Interestingly, both systemic atherosclerosis and lower vitamin D levels are associated with cardiovascular disease and increased mortality [30]. Therefore, participants with UUI may be at high risk of death from cardiovascular disease. Contrary to death, there was no association between UUI and the occurrence of LOI. Although this study did not clarify the mechanisms, some hypothetical explanations can be considered. First, there might be a difference between diseases that cause LOI and those that cause death among older adults with UUI. Although UUI is associated with falls and fractures [10,11], UUI may not be associated with other conditions associated with a risk of LOI, such as dementia, osteoporosis, or arthritis. Based on this explanation, UUI might be associated with death, not LOI, among participants in our study. Second, limited medical accessibility may be a relevant factor. The towns of Minami-Aizu and Tadami are both located in the remote countryside in Fukushima and residents have very limited medical resources. Furthermore, UUI may be a high-risk factor for cardiovascular disease. Therefore, we must consider that UUI may be associated with cardiovascular deaths without LOI among residents living in rural areas with limited accessibility to tertiary hospitals [31].

The limitations of this study were as follows: first, this was a population-based study in Japan; thus, the results are only generalizable to rural Japanese participants. Consequently, it is unclear whether these results apply to older adults living in urban areas of Japan or in other

A

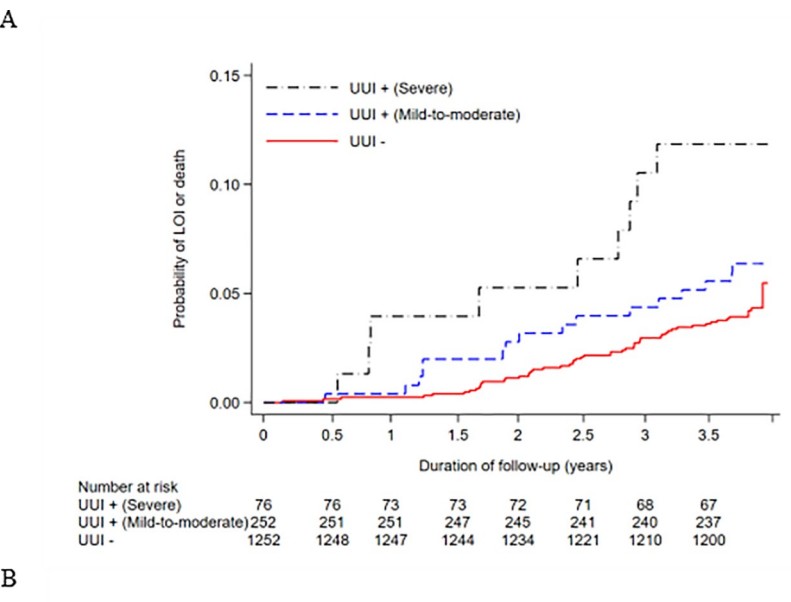

B

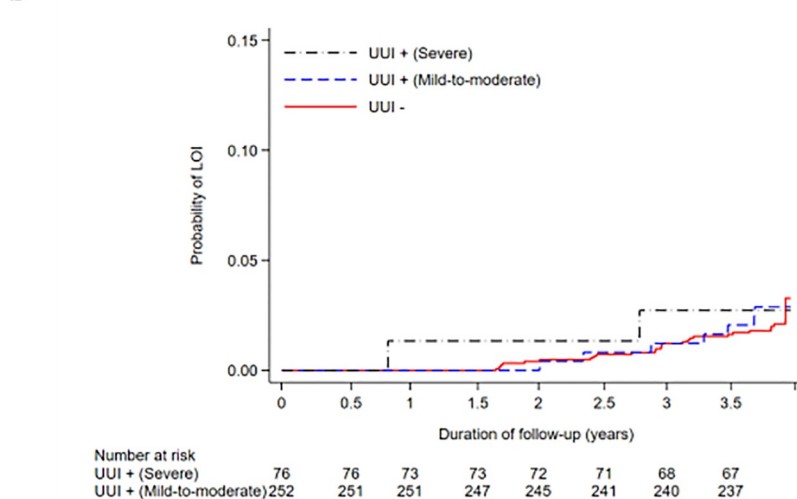

C

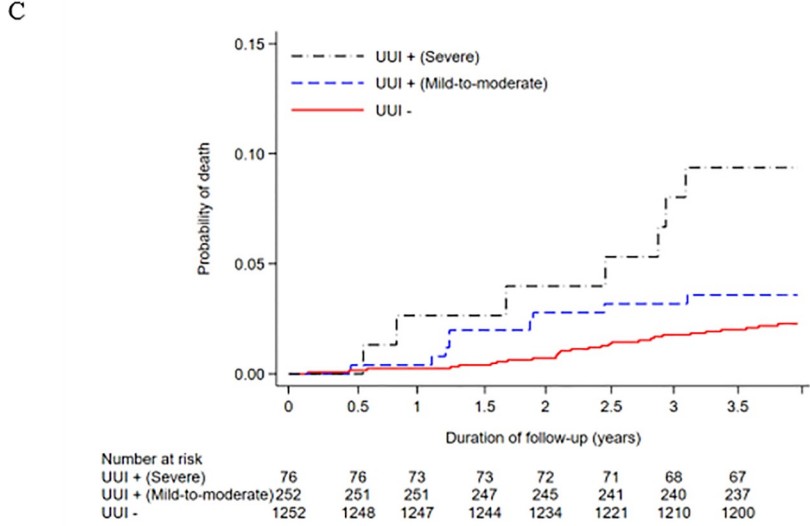

**Fig 3. Survival curve describing the association between no/mild-to-moderate/severe UUI and the outcomes using Kaplan–Meier method.** A, LOI or death; B, LOI; C, death.

countries. Second, LTCI application requires individuals' self-declaration; therefore, those with LOI may not always be LTCI certified. Third, the observation period in this study was relatively short: a maximum of four years. Consequently, we only observed 35 LOIs and 44 deaths; therefore, further studies with larger sample sizes and longer observation periods are warranted. Fourth, this study did not measure bladder medication; therefore, misclassification of the presence of UUI, which affects the statistical estimation towards the null, might exist. Fifth, this study may have unmeasured residual confounders because of the nature of observational research. For example, this study did not adjust for the use of diuretics which may worsen UUI symptoms and might therefore be a potential confounding factor [32]. Furthermore, this study did not investigate cognitive function, which is an important factor in UUI, LOI, and death among older adults [33,34]. Further studies that address the aforementioned limitations are needed.

The strengths of the study include the involvement of independent older adults living in communities, which is an important population for studies related to healthy aging and longevity [35]. We also adjusted for important potential confounders that were lacking in other studies [9], including participants' characteristics (age, sex, BMI, smoking status, and alcohol use) and comorbidities (hypertension, dyslipidemia, diabetes mellitus, heart disease, and stroke). Additionally, there were only seven missing data points related to exposure and we conducted follow-up studies with participants (1,572/1,580 = 99.5%). The follow-up analysis allowed for the vast majority of outcomes to be accurately measured. Finally, sensitivity analyses were performed to confirm statistical robustness.

**Table 3. Sensitivity analysis: multivariable Cox proportional model estimating HR for the occurrence of LOI and death.**

| | LOI | | | Death | | | |
|---|---|---|---|---|---|---|---|
| | HR | 95% CI | | HR | 95% CI | | |
| Presence of UUI (ref. no UUI) | | | | | | | |
| Model 1 | 1.34 | 0.63 | – 2.86 | **2.22** | **1.20** | **–** | **4.11** |
| Model 2 | 1.09 | 0.51 | – 2.33 | **2.19** | **1.17** | **–** | **4.10** |
| Model 3 | 1.09 | 0.51 | – 2.36 | **2.24** | **1.20** | **–** | **4.21** |
| Model 4 | 1.07 | 0.49 | – 2.33 | **2.29** | **1.22** | **–** | **4.31** |
| Presence of mild-to-moderate UUI (ref. no UUI) | | | | | | | |
| Model 1 | 1.35 | 0.58 | – 3.10 | 1.62 | 0.76 | – | 3.42 |
| Model 2 | 1.11 | 0.48 | – 2.57 | 1.63 | 0.76 | – | 3.49 |
| Model 3 | 1.11 | 0.48 | – 2.59 | 1.68 | 0.79 | – | 3.61 |
| Model 4 | 1.08 | 0.46 | – 2.54 | 1.69 | 0.78 | – | 3.64 |
| Presence of severe UUI (ref. no UUI) | | | | | | | |
| Model 1 | 1.32 | 0.31 | – 5.55 | **4.29** | **1.87** | **–** | **9.82** |
| Model 2 | 1.01 | 0.24 | – 4.28 | **3.55** | **1.57** | **–** | **8.01** |
| Model 3 | 1.03 | 0.24 | – 4.40 | **3.96** | **1.70** | **–** | **9.24** |
| Model 4 | 1.05 | 0.24 | – 4.49 | **4.18** | **1.78** | **–** | **9.79** |

Note. LOI: loss of independence. HR: hazard ratio. CI: confidence intervals. UUI: urgency urinary incontinence. BMI: body mass index.

**Model 1**—crude model. **Model 2**—adjusted for age and gender. **Model 3**—adjusted for body mass index, smoking status, and alcohol use, in addition to model 2. **Model 4**—adjusted for hypertension, dyslipidemia, diabetes mellitus, and histories of heart disease and stroke, in addition to model 3.

## Conclusion

This study found an association between UUI and the occurrence of LOI or death as well as death alone, but no association with LOI alone among community-dwelling older adults living independently in Japan. Our results suggest that there may be a difference between diseases that cause LOI and those that cause death among community-dwelling older adults with UUI. Our results are based on limited sample size in a rural area in Japan; therefore, further studies conducted with larger sample sizes in different settings in Japan and other countries are still warranted.

## Supporting information

**S1 Fig. Description of log-minus-log-survival plots for the evaluation of proportional hazard assumption in the primary analysis.** A, LOI or death; B, LOI; C, death.
(DOCX)

**S2 Fig. Results of cumulative incidence for the competing analyses.** A, with or without UUI for LOI in the secondary analysis; B, with or without UUI for death in the secondary analysis; C, with mild-to-moderate UUI, severe UUI, and without UUI for LOI in the tertiary analysis; D, with mild-to-moderate UUI, severe UUI, and without UUI for death in the tertiary analysis.
(DOCX)

**S3 Fig. Description of log-minus-log-survival plots for the evaluation of proportional hazard assumption in the tertiary analysis.** A, LOI or death; B, LOI; C, death.
(DOCX)

**S1 Table. Baseline characteristics of participants classified into UUI severity.**
(DOCX)

**S2 Table. Results of competing risk regression analysis.**
(DOCX)

## Acknowledgments

We deeply appreciate the LOHAS participants, staff of the public offices in Tadami and Minami-Aizu, and LOHAS research members for supporting this study. We thank Editage (www.editage.jp) for English language editing.

## Author Contributions

**Conceptualization:** Takashi Yoshioka, Yosuke Yamamoto.

**Data curation:** Takashi Yoshioka, Yosuke Yamamoto.

**Formal analysis:** Takashi Yoshioka.

**Investigation:** Takashi Yoshioka, Tsukasa Kamitani, Kenji Omae, Sayaka Shimizu, Shunichi Fukuhara, Yosuke Yamamoto.

**Methodology:** Takashi Yoshioka, Tsukasa Kamitani, Kenji Omae, Sayaka Shimizu, Yosuke Yamamoto.

**Project administration:** Tsukasa Kamitani, Shunichi Fukuhara, Yosuke Yamamoto.

**Resources:** Tsukasa Kamitani, Yosuke Yamamoto.

**Software:** Takashi Yoshioka.

**Supervision:** Tsukasa Kamitani, Kenji Omae, Sayaka Shimizu, Shunichi Fukuhara, Yosuke Yamamoto.

**Validation:** Takashi Yoshioka, Tsukasa Kamitani, Kenji Omae, Sayaka Shimizu, Shunichi Fukuhara, Yosuke Yamamoto.

**Visualization:** Takashi Yoshioka, Yosuke Yamamoto.

**Writing – original draft:** Takashi Yoshioka.

**Writing – review & editing:** Tsukasa Kamitani, Kenji Omae, Sayaka Shimizu, Shunichi Fukuhara, Yosuke Yamamoto.

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
