## [Decision Letter · Decision Letter 0]

8 Dec 2020

PONE-D-20-34134

Urgency urinary incontinence, loss of independence, and increased mortality in older adults: A cohort study

PLOS ONE

Dear Dr. YAMAMOTO,

Thank you for submitting your manuscript to PLOS ONE. After careful consideration, we feel that it has merit but does not fully meet PLOS ONE’s publication criteria as it currently stands. Therefore, we invite you to submit a revised version of the manuscript that addresses the points raised during the review process.

We look forward to receiving your revised manuscript.

Kind regards,

Pasquale Abete

Academic Editor

PLOS ONE

Journal Requirements:

2.) We note that you have indicated that data from this study are available upon request. PLOS only allows data to be available upon request if there are legal or ethical restrictions on sharing data publicly. For information on unacceptable data access restrictions, please see http://journals.plos.org/plosone/s/data-availability#loc-unacceptable-data-access-restrictions.

Additional Editor Comments (if provided):

According to Reviewer's comments the manuscript needs a minor revision.

Reviewers' comments:

Reviewer's Responses to Questions

**Comments to the Author**

1. Is the manuscript technically sound, and do the data support the conclusions?

Reviewer #1: Yes

Reviewer #2: Yes

2. Has the statistical analysis been performed appropriately and rigorously? 

Reviewer #1: Yes

Reviewer #2: Yes

3. Have the authors made all data underlying the findings in their manuscript fully available?

Reviewer #1: Yes

Reviewer #2: Yes

4. Is the manuscript presented in an intelligible fashion and written in standard English?

Reviewer #1: Yes

Reviewer #2: Yes

5. Review Comments to the Author

Reviewer #1: To the author: I delighted in reading the paper and I found the subject very interesting. It is well structured and written, however some aspects could be more extensively argued in a minor revision.

Minor revision

Method: Some medications, such as diuretics, could worsen urinary urge and incontinence effect in older adults. In this manuscript ten confounding covariates were considered with the exception of drugs. Therefore a difference between the two groups of subjects with and without urgency urinary incontinence in terms of medications is not explored. This could fade the relevance of some questionnaire items (How often do you urinate because you cannot defer the sudden desire to urinate? “never, less than once a week, once a week or more, about once a day, 2–4 times a day, and 5 times a day or more”). If data on medications lacking, the authors could debate the subject in the discussion and in the limitation of the study.

Discussion: The majority of older adults with urinary incontinence present nocturnal lower urinary tract symptoms. Nocturia as well as urinary icontinence in older people is associated with high morbidity and mortality. This aspect could be mentioned in the discussion session. Please see references Denys MA et al. Int J Urol. 2017 and Galizia et al. J Am Med Dir Assoc. 2012.

Reviewer #2: The Authors investigated the longitudinal association of urgency urinary incontinence with loss of independence (LOI) or death among independent community dwelling older adults. Data derived from a population-based cohort study in Japan in which 1,580 participants aged ≥65 years were enrolled in 2010. Urgency urinary incontinence (UUI) was measured by a questionnaire based on the definition of UUI from the International Continence Society. The primary outcome was defined as incidence of LOI or death. After the check-up in 2010, the outcome was monitored until March 2014. A multivariable Cox proportional hazard analysis was performed to estimate the hazard ratio for the outcome and for LOI and death separately. Ten potential confounders were adjusted in the analysis. Among all participants, 328 reported UUI. The incidence rates of the outcome were 20.4 and 11.4 (per 1,000 person–years) among participants with and without UUI, respectively. After multivariable adjustment, those who experienced UUI showed a substantial association with LOI or death (HR, 1.65; 95% CI, 1.01–2.68). However, they did not show such an association with LOI alone (HR, 1.07; 95% CI, 0.49–2.33). On the other hand, those with UUI exhibited a substantial association with death (HR, 2.23; 95% CI, 1.22–4.31). I found this study of interest, well designed, and conducted. Only one point of criticism is the lack of cognitive status evaluation and incident disability. Please, consider to cite this reference: Solfrizzi V, Scafato E, Lozupone M, Seripa D, Giannini M, Sardone R,Bonfiglio C, Abbrescia DI, Galluzzo L, Gandin C, Baldereschi M, Di Carlo A, Inzitari D, Daniele A, Sabbà C, Logroscino G, Panza F; Italian Longitudinal Study on Aging Working Group. Additive Role of a Potentially Reversible Cognitive Frailty Model and Inflammatory State on the Risk of Disability: The Italian Longitudinal Study on Aging. Am J Geriatr Psychiatry. 2017 Nov;25(11):1236-1248.

6. PLOS authors have the option to publish the peer review history of their article (what does this mean?). If published, this will include your full peer review and any attached files.

Reviewer #1: No

Reviewer #2: No

---

## [Author Response · Author response to Decision Letter 0]

18 Dec 2020

Comments from Reviewer #1 

Comment: 

To the author: I delighted in reading the paper and I found the subject very interesting. It is well structured and written, however some aspects could be more extensively argued in a minor revision.

Response: 

Thank you very much. We carefully designed and conducted this research. We have changed the manuscript to reflect the suggestions provided by the reviewers. 

Comment 1:

Method: Some medications, such as diuretics, could worsen urinary urge and incontinence effect in older adults. In this manuscript ten confounding covariates were considered with the exception of drugs. Therefore a difference between the two groups of subjects with and without urgency urinary incontinence in terms of medications is not explored. This could fade the relevance of some questionnaire items (How often do you urinate because you cannot defer the sudden desire to urinate? “never, less than once a week, once a week or more, about once a day, 2–4 times a day, and 5 times a day or more”). If data on medications lacking, the authors could debate the subject in the discussion and in the limitation of the study.

Response:

Thank you very much for the insightful comment. We agree that diuretics could worsen UUI. Contrary to anticholinergics (described as “bladder medication” in the manuscript), use of diuretics may be a confounding factor. This might be considered as another limitation regarding residual and unmeasured confounding factors in our study. As such, we added some sentences and a reference to the Discussion.

Changes:

Lines 299–302

Fifth, this study may have unmeasured residual confounders because of the nature of observational research. For example, this study did not adjust for the use of diuretics, which may worsen UUI symptoms and might therefore be a potential confounding factor [32]. 

[32] Ekundayo OJ, Markland A, Lefante C, Sui X, Goode PS, Allman RM, Ali M, Wahle C, Thornton PL, Ahmed A. Association of diuretic use and overactive bladder syndrome in older adults: a propensity score analysis. Arch Gerontol Geriatr. 2009;49(1): 64–68. https://doi.org/10.1016/j.archger.2008.05.002. 

Comment 2:

Discussion: The majority of older adults with urinary incontinence present nocturnal lower urinary tract symptoms. Nocturia as well as urinary incontinence in older people is associated with high morbidity and mortality. This aspect could be mentioned in the discussion session. Please see references Denys MA et al. Int J Urol. 2017 and Galizia et al. J Am Med Dir Assoc. 2012.

Response:

We entirely agree that nocturia is another important urinary symptom associated with LOI or death among older adults. Following the reviewer’s suggestion, we added text and a reference regarding nocturia to the Discussion.

Changes:

Lines 245–247

Although several previous studies revealed an association between nocturia and LOI or death [21, 22], it was unclear whether such an association was present for UUI. For example, some studies showed that UUI (not SUI) affected falls, fractures, and mobility impairment, [10,11] evidence illustrating the association between UUI and LOI is scarce.

[21] Denys MA, Decalf V, Kumps C, Petrovic M, Goessaert AS, Everaert K. Pathophysiology of nocturnal lower urinary tract symptoms in older patients with urinary incontinence. Int J Urol. 2017;24(11): 808–815. https://doi.org/10.1111/iju.13431.

[22] Galizia G, Langellotto A, Cacciatore F, Mazzella F, Testa G, Della-Morte D et al. Association between nocturia and falls-related long-term mortality risk in the elderly. J Am Med Dir Assoc. 2012;13(7): 640–644. https://doi.org/10.1016/j.jamda.2012.05.016.

Comments from Reviewer #2

Comment: 

The Authors investigated the longitudinal association of urgency urinary incontinence with loss of independence (LOI) or death among independent community dwelling older adults. Data derived from a population-based cohort study in Japan in which 1,580 participants aged ≥65 years were enrolled in 2010. Urgency urinary incontinence (UUI) was measured by a questionnaire based on the definition of UUI from the International Continence Society. The primary outcome was defined as incidence of LOI or death. After the check-up in 2010, the outcome was monitored until March 2014. A multivariable Cox proportional hazard analysis was performed to estimate the hazard ratio for the outcome and for LOI and death separately. Ten potential confounders were adjusted in the analysis. Among all participants, 328 reported UUI. The incidence rates of the outcome were 20.4 and 11.4 (per 1,000 person–years) among participants with and without UUI, respectively. After multivariable adjustment, those who experienced UUI showed a substantial association with LOI or death (HR, 1.65; 95% CI, 1.01–2.68). However, they did not show such an association with LOI alone (HR, 1.07; 95% CI, 0.49–2.33). On the other hand, those with UUI exhibited a substantial association with death (HR, 2.23; 95% CI, 1.22–4.31). I found this study of interest, well designed, and conducted. 

Response:

We thank the reviewer for this precise summary of our study and the positive comments. 

Comment 1:

Only one point of criticism is the lack of cognitive status evaluation and incident disability. Please, consider to cite this reference: Solfrizzi V, Scafato E, Lozupone M, Seripa D, Giannini M, Sardone R,Bonfiglio C, Abbrescia DI, Galluzzo L, Gandin C, Baldereschi M, Di Carlo A, Inzitari D, Daniele A, Sabbà C, Logroscino G, Panza F; Italian Longitudinal Study on Aging Working Group. Additive Role of a Potentially Reversible Cognitive Frailty Model and Inflammatory State on the Risk of Disability: The Italian Longitudinal Study on Aging. Am J Geriatr Psychiatry. 2017 Nov;25(11):1236-1248.

Response: We thank the reviewer for providing us with an important viewpoint and suggesting the related paper. We agree that cognitive impairment plays an important role in UI and its treatment. In addition, as the reviewer indicates, cognitive frailty is associated with LOI and death among older adults. Considering these points, we added text and references to the Discussion, including the reference suggested by the reviewer.

Changes: 

Lines 302–304

Furthermore, this study did not investigate cognitive function, which is an important factor in UUI, LOI, and death among older adults [33,34]. Further studies that address the aforementioned limitations are needed.

[33] Booth J, Kumlien S, Zang Y. Promoting urinary continence with older people: key issues for nurses. Int J Older People Nurs. 2009;4(1): 63–69. https://doi.org/10.1111/j.1748-3743.2008.00159.x. 

[34] Solfrizzi V, Scafato E, Lozupone M, Seripa D, Giannini M, Sardone R et al. Additive role of a potentially reversible cognitive frailty model and inflammatory state on the risk of disability: The Italian Longitudinal Study on Aging. Am J Geriatr Psychiatry. 2017;25(11): 1236–1248. https://doi.org/10.1016/j.jagp.2017.05.018.

In addition to the above comments, we have re-numbered the references following the insertion of new references.

---

## [Decision Letter · Decision Letter 1]

7 Jan 2021

Urgency urinary incontinence, loss of independence, and increased mortality in older adults: A cohort study

PONE-D-20-34134R1

Dear Dr. YAMAMOTO,

We’re pleased to inform you that your manuscript has been judged scientifically suitable for publication and will be formally accepted for publication once it meets all outstanding technical requirements.

Kind regards,

Pasquale Abete

Academic Editor

PLOS ONE

Additional Editor Comments (optional):

No further comments.

Reviewers' comments:

Reviewer's Responses to Questions

**Comments to the Author**

1. If the authors have adequately addressed your comments raised in a previous round of review and you feel that this manuscript is now acceptable for publication, you may indicate that here to bypass the “Comments to the Author” section, enter your conflict of interest statement in the “Confidential to Editor” section, and submit your "Accept" recommendation.

Reviewer #1: All comments have been addressed

Reviewer #2: All comments have been addressed

2. Is the manuscript technically sound, and do the data support the conclusions?

Reviewer #1: Yes

Reviewer #2: Yes

3. Has the statistical analysis been performed appropriately and rigorously? 

Reviewer #1: Yes

Reviewer #2: Yes

4. Have the authors made all data underlying the findings in their manuscript fully available?

Reviewer #1: Yes

Reviewer #2: Yes

5. Is the manuscript presented in an intelligible fashion and written in standard English?

Reviewer #1: Yes

Reviewer #2: Yes

6. Review Comments to the Author

Reviewer #1: All comments have been well addressed and the manuscript is now worthy to be considered for publications

Reviewer #2: I have no more suggestions to improve the manuscript that i find suitable for publication in the present form

7. PLOS authors have the option to publish the peer review history of their article (what does this mean?). If published, this will include your full peer review and any attached files.

Reviewer #1: No

Reviewer #2: No

---

## [Editor Report · Acceptance letter]

11 Jan 2021

PONE-D-20-34134R1 

Urgency urinary incontinence, loss of independence, and increased mortality in older adults: A cohort study 

Dear Dr. Yamamoto:

I'm pleased to inform you that your manuscript has been deemed suitable for publication in PLOS ONE. Congratulations! Your manuscript is now with our production department. 

Kind regards, 

on behalf of

Prof. Pasquale Abete 

Academic Editor

PLOS ONE